# Numerical Simulation Analysis of the Formation and Morphological Evolution of Asymmetric Crescentic Dunes

**Huiwen Zhang [1,2,3], Changlong Li [1], Jianhui Zhang [1], Zhen Wu [4,*], Zhiping Zhang [3], Jing Hu [1], Lei Cao [3], Longlong Song [3], Jianping Ma [3] and Bin Xiao [3]**

[1] National Forest Germplasm Resource Bank of Desert Plants in Minqin Desert Control Station of Gansu Province, Gansu Desert Control Research Institute, Lanzhou 733000, China; zhanghuiwen@lzb.ac.cn (H.Z.); changlongdune@163.com (C.L.); jianhuidune@163.com (J.Z.); hujingdune@163.com (J.H.)

[2] Postdoctoral Research Workstation, Gansu Desert Control Research Institute, Lanzhou 730070, China

[3] State Key Laboratory Breeding Base of Desertification and Aeolian Sand Disaster Combating, Gansu Desert Control Research Institute, Lanzhou 730070, China; zhipingdune@163.com (Z.Z.); caoleidune@163.com (L.C.); songlonglongdune@163.com (L.S.); jianpingdune@163.com (J.M.); xiaobindune@163.com (B.X.)

[4] Lanzhou Institute of Seismology of China Earthquake Administration, Earthquake Administration of Gansu Province, Lanzhou 730000, China

[*] Correspondence: wuzhen@lzb.ac.cn; Tel.: +86-139-1982-0160

**Abstract:** Generally, typical crescentic dunes in the ideal state are symmetrical, but it is difficult to form crescentic dunes with two perfectly symmetrical horns under actual conditions. Among many environmental factors, bidirectional winds, the size of sand particles, topography, epiphyte vegetation, and dune collision are important reasons for the asymmetric evolution of sand dunes. Few existing studies have revealed the mechanism of the morphological evolution of asymmetric crescentic dunes, especially in regard to the role of wind in a complex dune's morphology. In this study, we used the Reynolds Averaged Navier-Stokes (RANS) and mass balance models to simulate the asymmetric forms and flow fields of crescentic dunes and analyzed the potential causes of the asymmetry among the above aspects. The results showed that: (1) the angle of the bidirectional winds significantly changed the structure of vortices around the sand dune; (2) for crescentic dunes with coarser sand, the deposit continuity was better, the extension of the single horn was maintained for a long time, and the extended horn took longer to die out; (3) the crescentic dune deformed according to the direction of the inclination of the terrain, and the shear stress of a dune on a slope was related to the slope, width, or height; (4) whether there was epiphytic vegetation on a dune's surface had a great impact on the dune's migration; (5) the collision position of two dunes determined the shape of the two dunes after fusion. The simulation results indicated that the spatial–temporal differences in sand flux, caused by changes in flow fields that were induced by various factors, determined the evolutionary shape of crescentic dunes. These results can provide a reference for the study of the erosion of surface flow fields on various dunes and for the prevention and control of wind and sand disasters in the Gobi Desert area.

**Keywords:** asymmetric crescentic dunes; morphological evolution; surface flow field; numerical simulation

## 1. Introduction

Sand dunes form in response to the flow-induced mobilization of sediment [1–4]. As is commonly known, crescentic dunes are formed mainly under unidirectional winds [5,6]. However, a large number of asymmetric crescentic dunes are widespread in the Gobi Desert area, indicating that there are many factors leading to the asymmetric characteristics of crescentic dunes [7–10]. Therefore, it is of great significance to study the morphological evolution of crescentic dunes, to analyze their regional background in the inland region of northwest China, to reveal their morphological characteristics and distribution pattern, to compare the differences among the causes and influencing factors of these dunes, and

to deeply understand the characteristics, influencing factors, and action modes of sand deposition landforms [11,12].

Because conventional field observations are time-consuming, costly, and long-cycled, the numerical simulation, a time- and labor-saving method, has begun to be widely used in the study of dune changes [13–15]. A numerical model can transform the environmental factors into functional relations and adjust the parameters according to the actual environmental conditions in the field. Then, the results that are driven by the theoretical equation can be reflected in the model. Thus, many simulation results that are highly suited to the actual situation but would be difficult to measure can be obtained [16–19]. Sauermann derived a minimal phenomenological continuous saltation model for sand transport and geomorphological applications that could simulate the morphologic changes in sand dunes with time [20]. Another such model has been developed during the course of the last decade [21–23], commonly referred to as the Exner equation, and has been presented and used in various forms for the analysis of earth surface morphodynamics [24,25]. This model assumed that the air flow varied exponentially with height, but the transport process of sand particles took place in the lowest atmosphere in a highly turbulent regime, so the simulation of air flow could not be more realistic. Then, a more accurate fluid dynamics model was presented that required solving a set of conservation equations for air, usually restricted in terms of mass and momentum balance. When the wind was treated as a one-constituent incompressible viscous fluid, the classical Navier–Stokes equations for incompressible flow were obtained [26]. Consequently, to describe turbulent flow, two different approaches are mainly used, Reynolds-averaged Navier–Stokes equations (RANS) [27,28] and large-eddy simulation (LES) [29–31]. These models typically describe the transport of local substances through partial differential equations [32,33]. Moreover, these models require the users to have sufficient mathematical skills.

As early as the last century, researchers carried out studies of the evolution of crescentic dunes under the action of bidirectional winds [34,35]. According to Bagnold's conceptual model, bidirectional winds with different velocities could promote the development of asymmetric crescentic dunes, while those with the same velocities caused crescentic dunes to evolve into Seif dunes [36]. Some field observational phenomena supported the results of this model [8,37]. These patterns represent both the external environmental conditions under which the fields evolved and the internal dynamics of the self-organization of the dunes [38]. This type of dune is more complex because of sediment availability, which controls the volumetric scales that dunes can reach. Furthermore, the presence of vegetation absorbs aerodynamic momentum fluxes and inhibits the morphodynamics of dunes by suppressing the modes of sand flux [9,39,40]. The formation and evolution of crescentic dunes are influenced by many other factors besides the above, such as topography, the size of sand particles, and so on [41–44]. As a result of these factors, classical symmetry in crescentic dunes is very rare. The degree of extension varies greatly, with the extension horn being either straight or curved [45]. Many crescentic dunes even shrink in size during migration, finally resulting in the formation of several small sand mounds around them [33,46].

To analyze the causes and mechanisms of the morphological changes in asymmetric crescentic dunes, we simultaneously used the Exner and RANS models for the description of mass migration to simulate the wind flow and evolution of asymmetric crescentic dunes and expounded the deep reasons for the changes in sand flux on both horns of dunes, caused by various factors. The main objectives of our study included: (1) simulating the morphological evolution of asymmetric dunes based on different factors; (2) analyzing the causes of single horn extension based on the simulation results; (3) evaluating the practicability and effectiveness of the RANS and continuity equations; (4) explaining and revealing the underlying causes of asymmetry in crescentic dunes. The specific methods and results are described below.

## 2. Research Methods and Parameter Setting

### 2.1. Model of Dunes

Modelling the formation and morphological changes of crescentic dunes needs to consider the following factors: the influence of topography on wind, the amount of sand moving by wind, the erosion of dunes by wind, and the deposition and movement of sand particles. In our study, the sand bed was considered to be immobile, and the sand dune was a thin fluid-like granular layer on the sand bed. When the model begins to calculate the wind erosion of dunes, the elevation of the different positions of the dunes is updated repeatedly with the changes of the mass balance, and then the new morphology of dunes is continuously obtained. The specific model settings are shown as follows.

### 2.2. Simulation of Shear Stress on Dunes' Surface by Wind

In general, the velocity of the wind field of a unidirectional wind on the surface of dunes conforms to the logarithmic distribution with the distribution of the height—that is, to conform to the law of logarithmic distribution of the velocity of Plant von Kármán, derived from the theory of mixed length of Plant:

$$u(z) = \frac{u_0}{k} \ln(\frac{z}{z_0}) \tag{1}$$

where $k$ is the constant of von Kármán and its value is set as 0.375; $u_0$ is the friction velocity of wind and its formula is $u_0 = \frac{\sqrt{\tau}}{\rho}$; $u(z)$ is the shear velocity of wind at the height of $z$; $\tau$ is the shear stress of the dune's surface, and its specific form can be expressed as follows:

$$\tau_0 = \rho_{fluid} |u_0| u_0 \tag{2}$$

where $\rho$ is the density of the air, $z$ is the roughness of the dune's surface, and its value is determined by the size of the sand particles on the sand bed.

For the simulation of the shear velocity of sand particles, we referred to the description formula of the transmission velocity of saturated sand [21] and some empirical observations [47]. In this study, we supposed that the height of the saturated sand layer was $z_m$, and its value was about 20 mm; the value of $z_0$ was about 10 μm; and the value of $z_1$ was about 3 mm. Then, the effective wind velocity can be written as follows:

$$v_{eff}(x, y) \approx \frac{u_t}{k} \left[ \ln \frac{z_1}{z_0} + \frac{z_1}{z_m} \left\{ \frac{u(x, y)}{u_{threshold}} - 1 \right\} \right] \tag{3}$$

The relation between the critical velocity and the size of the sand particles when the sand begins to move can be expressed as follows:

$$u_t = \xi \sqrt{\frac{\rho_s - \rho}{\rho} g d} \tag{4}$$

where $u_t$ is the friction velocity; $\rho$ is the density of the sand particles; $d$ is the size of the sand particles; $g$ is the acceleration of gravity; and $\xi$ is the empirical coefficient, and its value is set as 0.1.

The velocity of sand particles can be expressed by the movement of sand particles ($u_s$) in the horizontal direction at different heights of $z_1$, which mainly refers to the splash that is caused by the equilibrium action of sand particles' momentum and the acceleration of gravity. It represents the velocity of a saturated layer rather than the velocity of an individual sand particle.

$$\frac{(\vec{v}_{eff} - \vec{u}_s) \left| \vec{v}_{eff} - \vec{u}_s \right|}{u_i^2} - \frac{\vec{u}_s}{2\alpha \left| \vec{u}_s \right|} - \vec{\nabla} h = 0 \tag{5}$$

where $\vec{u}_s \approx \left(v_{eff} - \frac{u_f}{\sqrt{2\alpha A}}\right)\vec{e}_\tau - \frac{\sqrt{2\alpha}u_f}{A}\vec{\nabla}h$ and $A = \left|\vec{e}_\tau + 2\alpha\vec{\nabla}h\right|$.

### 2.3. Relation between Starting Velocity of Wind and Topographic Angle

The influence of ground surface properties on wind velocity is that rough surface must affect the starting wind velocity because of its large friction resistance. In addition, due to the surface slope, the starting velocity of wind in the sand particles increases at the windward slope, and decreases at the leeward slope. Thus, we used Howard's quantitative description [37] of the starting wind velocity and surface slope:

$$u_t = F^2 \cdot d \cdot [(\tan^2 a * \cos^2\theta - \sin^2\chi * \sin^2\theta^{\frac{1}{2}}) - \cos\chi * \sin\theta] \tag{6}$$

where $F = B(g\rho\_s/\rho)^{1/2}$, $B$ is a dimensionless constant with a value of 0.31. $d$ is the size of the sand particles, $\alpha$ is the internal friction angle, $\theta$ is the slope of the ground, and $\chi$ is the angle between the wind direction and the normal line of the sand bed.

### 2.4. Simulation Methods of Sand Flux

For the simulation of sand flux, we used the nonlinear transport equation [20] to describe the spatial variation of saturated sand flow:

$$\frac{\partial q}{\partial x} = \frac{q}{l_s}\left(1 - \frac{q}{q_s}\right) \tag{7}$$

where $l_s$ is the maximum length and $q_s$ is the amount of saturated sand in the air. They are described, respectively, as follows:

$$\vec{q}_s(u) = \frac{2\alpha}{g}\frac{\rho}{\rho_{sand}}(u^2 - u_{threshold}^2)\vec{u}_s \tag{8}$$

$$l_s(u) = \frac{2\alpha\left|\vec{u}_s\right|^2}{\gamma g}\frac{1}{(u/u_{threshold})^2 - 1} \tag{9}$$

where $\alpha$ is the effective recovery coefficients [21] and $\gamma$ is the description parameter of the splash process [20].

### 2.5. Evolution of Dune's Morphology

The scale of sand flux mainly depends on the wind shear stress on the dune's surface. The greater the shear stress, the greater the sand flux. Although wind in different directions and scales can lead to shear stresses with different scales, according to previous studies, there is a threshold for the velocity of the shear stress of wind on the dune's surface [34]—that is, the maximum amount of sand particles that the wind can carry. We fitted the relationship between the maximum velocity and the size of the sand particles according to the previous calculation results on the threshold of velocity [48]. The specific formula is shown as follows:

$$u_{threshold} = 1.76 * 10^{-16} * S_d^2 + 0.3455 * S_d + 0.1155 \tag{10}$$

Thus, we used the Exner equation to describe sand flux in our study [24,49].

$$\frac{\partial h}{\partial t} + \nabla_{(xi,yi,zi)} \cdot \vec{q} = 0 \tag{11}$$

where $q$ is a function of the relation between the transmission rate and the shear stress. Since the shear stress is related to the velocity of wind, $q$ can be expressed as follows:

$$q_s = \varepsilon\sqrt{(s-1)gd_{50}^3} \cdot \left(\frac{4\tau(u)}{\rho_f(s-1)gd_{75}} - \tau_c\right)^{3/2} \tag{12}$$

$$\tau(u) = 1/8(\rho_f f |u|^2) \tag{13}$$

where $\rho$ denotes the density of sediment, $d_{75}$ is the median size of the sand particles (3 mm), and $\tau_c$ is the dimensionless critical shear stress (0.05).

In combination with Equations (12) and (13), Equation (11) can be used to describe the loss of sand particles and the quantitative changes in the deposition of the sand particles under shear stress.

To simplify the description of Equation (11), the change of sediment morphology on the surface of the dune can be regarded as the movement of continuous fluid under the action of external force. Then, the dune will change in sand flow in all three directions. If the sand flow is taken as the amount of mass lost, an equilibrium equation can be obtained as follows:

$$\frac{\partial h}{\partial t} + u\frac{\partial s}{\partial x} + v\frac{\partial s}{\partial y} - w = 0 \tag{14}$$

With a given accumulation function, $s$ is the dune's surface height and $u$, $v$, $w$ are the sand fluxes in different directions, respectively.

In addition to considering the migration of the dune, the avalanche phenomenon on the surface during evolution can be explained by the slope angle. This avalanche phenomenon is closely related to the height of the sand dune. We used a diffusion equation combined with a mass balance equation to calculate the mass accumulation. The diffusion equation can be written as follows:

$$q_{ava} = -\rho_s \varepsilon \nabla h \tag{15}$$

where $\varepsilon$ is a adjustment factor, and its value is related to the change rate of the height of the dune and the particle sizes of the model. It is also necessary to define the collapse angle as $\theta_a$ According to the previous research methods [33], we set that when the slope of the dune's surface exceeded 34° (that is, when $\theta_a < \theta_{rep}$), there would be no avalanche or slippage movement. In this case, $q_{ava} = 0$, otherwise, the sand particles would be redistributed according to the sand flow:

$$\varepsilon = \begin{cases} \dfrac{E\left[tanh(|\nabla h|) - (tan\theta_{rep})\right]}{|\nabla h|\rho_s} & |\nabla h| > tan\theta_{rep} \\ 0 & |\nabla h| \leq tan\theta_{rep} \end{cases} \tag{16}$$

For a sufficiently large coefficient $E$, the slope is relaxed independently of $E$. The value of $E$ is set as $-0.9$. This condition first runs under a constant condition and finally reaches a stable state. Because of the different period of reconstruction that is caused by the slope of the leeward slope, the setting of the critical leeward slope greatly affects the shape of the longitudinal section of the whole dune. The specific model settings are as above, and the specific values of the parameters that are used in the model are shown in Table 1.

**Table 1.** Specific values of parameters used in the model.

| Symbol | Parameter | Value | Unit |
|---|---|---|---|
| $d$ | the size of sand particles | 250 | μm |
| $\rho_{fluid}$ | the density of the air | 1.225 | kg/m$^3$ |
| $\nu$ | the viscosity of the air | $1.75 \times 10^{-5}$ | kg/ms |

**Table 1.** *Cont.*

| Symbol | Parameter | Value | Unit |
|---|---|---|---|
| $\rho_{sand}$ | the density of sand particles | 2650 | kg/m$^3$ |
| $u_{thredhold}$ | the threshold for velocity of shear | 0.2 | m/s |
| $\theta_{dyn}$ | the collapse angle | 33 | degree |

*2.6. Simulation of Wind Flow Field*

Due to the complexity of turbulence in the wind field, we performed simulations using the RANS equation [28,50–52], which is the most suitable model for simulating the motion of the fluid. The method represents the various physical parameters of the fluid as stable pulsating values, and then the equations of the statistical mean physical quantities can be obtained and applied to our study, where the continuum and momentum equations are shown as follows:

$$\frac{\partial \rho}{\partial t} + \frac{\partial}{\partial x_i}(\rho u_i) = 0 \tag{17}$$

$$v_x' \frac{\partial v_x'}{\partial x'} + v_y' \frac{\partial v_x'}{\partial y'} = -\frac{\partial p\prime}{\partial x'} + \frac{1}{\text{Re}}\left(\frac{\partial^2 v_x'}{\partial x'^2} + \frac{\partial^2 v_x'}{\partial y'^2}\right) \tag{18}$$

$$v_x' \frac{\partial v_y'}{\partial x'} + v_y' \frac{\partial v_y'}{\partial y'} = -\frac{\partial p\prime}{\partial x'} + \frac{1}{\text{Re}}\left(\frac{\partial^2 v_y'}{\partial x'^2} + \frac{\partial^2 v_y'}{\partial y'^2}\right) \tag{19}$$

where $\text{Re} = \frac{\rho D}{\nu}V_r$ , $D$ is the sizeof the sand particles, $v = 1.49 \times 10^{-5} \cdot \text{m}^2 \cdot \text{s}^{-1}$. $v$ represents the viscous coefficient of the air. $u$ represents the calculated mean velocity of wind. $V_r$ is the ratio and the relative velocity between the sand particle and the wind flow field. To obtain the accurate and stable results, a constant uniform wind field was used in this study.

*2.7. Interaction Force between Sand Particles and Wind Flow*

In general, moving sand particles are subjected to the drag force of air flow, electrostatic force, and gravity in the wind field. Among them, the drag force of air flow and the gravity of sand particles have the greatest influence on the motion of sand particles. This study ignored the influence of electrostatic force. For nearly spherical sand particles, their gravity, $F_g$, can be expressed as follows:

$$F_g = \frac{1}{6}\pi\rho_g D^3 g \tag{20}$$

where $\rho_g$ is the density of the sand particles, $g$ is the acceleration constant of gravity.

According to the research of Anderson and Haff [53], the drag force of the wind field on sand particles can be described as follows:

$$F_D = \frac{1}{8}C_D\rho_a\pi D^2|V_r|V_r \tag{21}$$

where $V_r$ is the relative velocity between the sand particles and the wind field, which can be expressed as follows:

$$V_r = \sqrt{(u - u_D) + (v - v_D)} \tag{22}$$

where $u_D$ and $v_D$ represent the velocity of sand particles in the horizontal and vertical directions, respectively. $C_D$ is the resistance coefficient, which can be calculated by the following empirical equation:

$$C_D = \left(0.63 + \frac{4.8}{\text{Re}^{\frac{1}{2}}}\right)^2 \tag{23}$$

where Re denotes the Reynolds coefficient. The fluid pressure gradient causes shear stress in the velocity direction, resulting in an up flow. It can be described as follows:

$$F_l = \frac{1}{8}\pi\rho_a C_l D^2 \left(u_{up}{}^2 - u_{down}{}^2\right) \tag{24}$$

where $u_{up}$ and $u_{down}$ are the velocity of the sand particles at the upper and lower boundary. $C_l$ is the coefficient of rising force, whose value is 0.85 times of $C_D$. Thus, the migration equation of the sand particles can be written as follows:

$$m_p \frac{dU_D}{dx} = F_g + F_D + F_l \tag{25}$$

where $m_p$ is the mass of the sand particles.

### 2.8. Validation of the Accuracy of the Simulation

To verify the reliability of the simulation results, we compared the measured velocity of wind and the actual simulated velocity of wind in the field at different locations on the main axis of a crescentic dune (Figure 1). The measured data were from a crescentic dune that was located in the northwest of Minqin County, China, which had a length of 6.50 m, a height of 0.37 m, and a maximum wind velocity of 11 m/s. The simulation conditions set the same length and height as measured, and the horns of the dune were set to a symmetrical shape. Although the local velocity of wind had some fluctuations, the velocity in the main direction of wind was relatively stable. Then, we simulated the variation of the horizontal velocity of wind with height at the windward toe, the leeward toe, and the different distances positions at the downwind. Figure 1a,b were the comparison between the simulation results and the measured data at the windward toe and the leeward toe. The two sets of data almost overlapped and had a good agreement, indicating that the RANS model could reproduce the actual distribution of wind velocity more truly, and the flow field on the dune's surface that was obtained from the simulation was credible. The simulation results of 5 m and 10 m after the leeward slope in Figure 1c,d were gradually different from the measured data, which indicated that with the increase in the distance between the observation points and the sand dune, the impact of the vortices on the wind velocity and wind direction gradually decreased, and there would be errors in the results of the simulation. This suggested that the closer the observation distance was to the sand dune, especially when the observation point was directly located on the surface of the sand dune, the more reliable were the wind velocity results that were simulated by the RANS model.

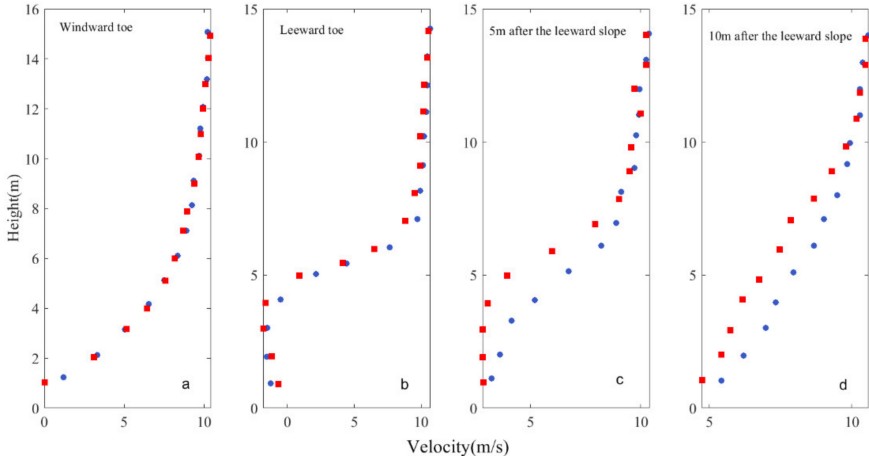

**Figure 1.** Comparison of the simulated and the actual measured velocity of wind in the field at different locations on the main axis of the sand dune (the round blue dots were the measured data, and the square red blocks were the simulation results).

## 3. Results

### 3.1. Simulation of the Effects of Bi-Directional Winds on Morphological Changes in Sand Dunes

In this study, we assumed that the wind velocities of the bi-directional winds were equal and both winds came from the same location, but the directions and the durations were different. The initial morphology of the dune was a nearly circular stacked sand dune. In the model, $T_{d1}$ was the duration of the main wind and $\tau_1$ was its shear stress. $T_{d2}$ was the duration of the secondary wind and $\tau_2$ was its shear stress, while $\theta$ was the angle between the two winds. Assuming that the bi-directional winds had the same shear stress on the dune's surface, the ratio of the duration was set to $r = T_{d2}/T_{d1}$. The duration of $T_{d1}$ was set to 1 and $r$ denoted the percentage of $T_{d2}$ to $T_{d1}$, where the main wind had a longer duration. To analyze the wind-induced changes in the morphology of dunes, the evolution of crescentic dunes was simulated with the same proportion of duration (r = 60%) and under the influence of bi-directional winds with different angles (Figure 2). The method was close to the previous research on bi-directional winds [54].

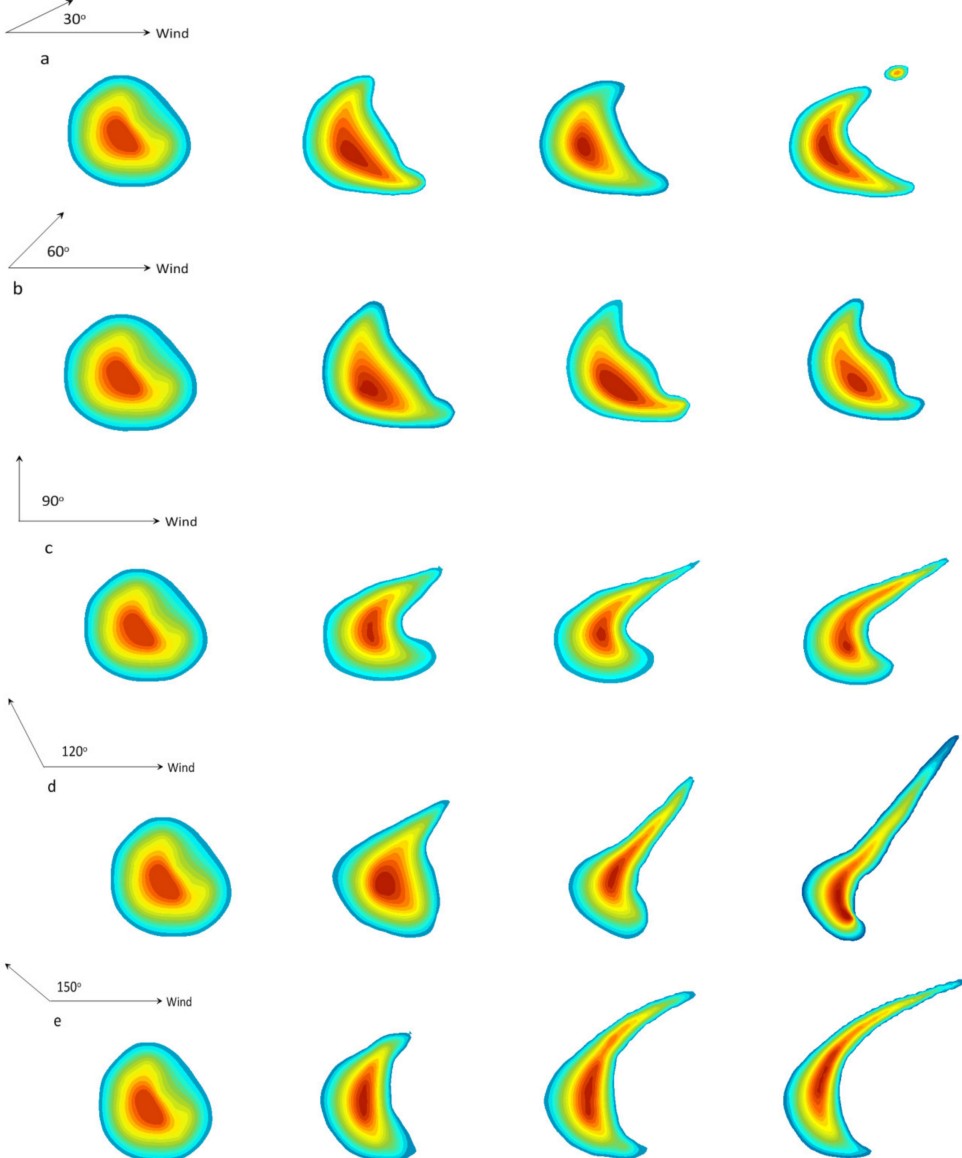

**Figure 2.** Changes in morphology of the dunes caused by bi-directional winds with different angles at r = 60% (the lateral form was state of changes over time). (**a**) At an angle of 30°; (**b**) at an angle of 60°; (**c**) at an angle of 90°; (**d**) at an angle of 120°; (**e**) at an angle of 150°.

Figure 2 showed that when the angles of the bi-directional winds were 30° and 60° (Figure 2a,b), the initial dune quickly developed the two horns and they extended more evenly. There was an expansion in the right horn, but it developed slowly. The windward toe gradually formed along the direction of the resultant force of the bi-directional winds. When the angle was 30°, the shape of the whole dune gradually accorded with the basic shape of the crescentic dune with the development of time. However, when the angle was 60°, although the whole dune still remained in the basic shape of the crescentic dune, it had actually been deformed and had a bulge that was formed by sand accumulation on the leeward slope. When the angle was 90° (Figure 2c), the dune exhibited an obvious one-horn extension during evolution, resulting in two-horn asymmetry; however, the length of the extended horn did not increase significantly over time. As the angle of the bi-directional winds was obtuse, the one-horn extension occurred obviously during the evolution, especially when the angle was 120° (Figure 2d).

Compared with the angle of 120°, the dune tended to be more slender and deformed into a linear dune at the angle of 150° (Figure 2e). Since the shear stress that was formed by the bi-directional winds was applied to both sides of the dune, so the shear stress in the direction of the resultant force was weakened. In this case, the dune was approximately an extruded body. Although the extension of the single horn occurred, the extension was slower because the combined force of the two winds was small, and the two forces offset each other. The above results suggested that the dunes were more prone to single-horn extension at larger divergence angles of bi-directional winds, which was consistent with previous studies [55].

### 3.2. Simulation of the Effects of Sand Particles' Size on Morphological Changes in Crescentic Dunes

Due to the gravity differences of sand particles, wind usually separates the sand particles of crescentic dunes, resulting in the regular distribution of sand particles with different size in different locations of dunes. There are two main distribution patterns in the size of the sand particles on the surface of dunes: one pattern is from the bottom to the top of the dune, in which the sand particles gradually become finer. The other pattern, also from the bottom to the top of the dune, involves the sand particles becoming coarser. Generally, the size distribution of the sand particles on dunes is closely related to the size composition of the sand particles in sand-sourced sediments. The migration of sand particles in crescentic dunes is similar to that of rolling transport. During the process of carrying sand, the wind automatically separates and distributes a certain size of the sand particles in a fixed position due to the weakened wind force with the distance [56]. Even if the size of the sand particles in the initial dune is not uniform, the wind separates and regularizes the new sand particles [47,57]. We simulated the two-dimensional migration trajectory of the sand particles with different sizes (Figure 3). Sand particles of different sizes marked indifferent colors on the graph represented the particles' suspension, jump, and creep. For the migration of sand dunes, the jump and creep of sand particles mainly constitute the morphological evolution of sand dunes, while suspense contributes little, due to its long and irregular migration distance. When the sand particles gather to a certain scale, it is often easy to form a small crescentic dune behind the main dune. The result that we obtained was in agreement with a previous study [58].

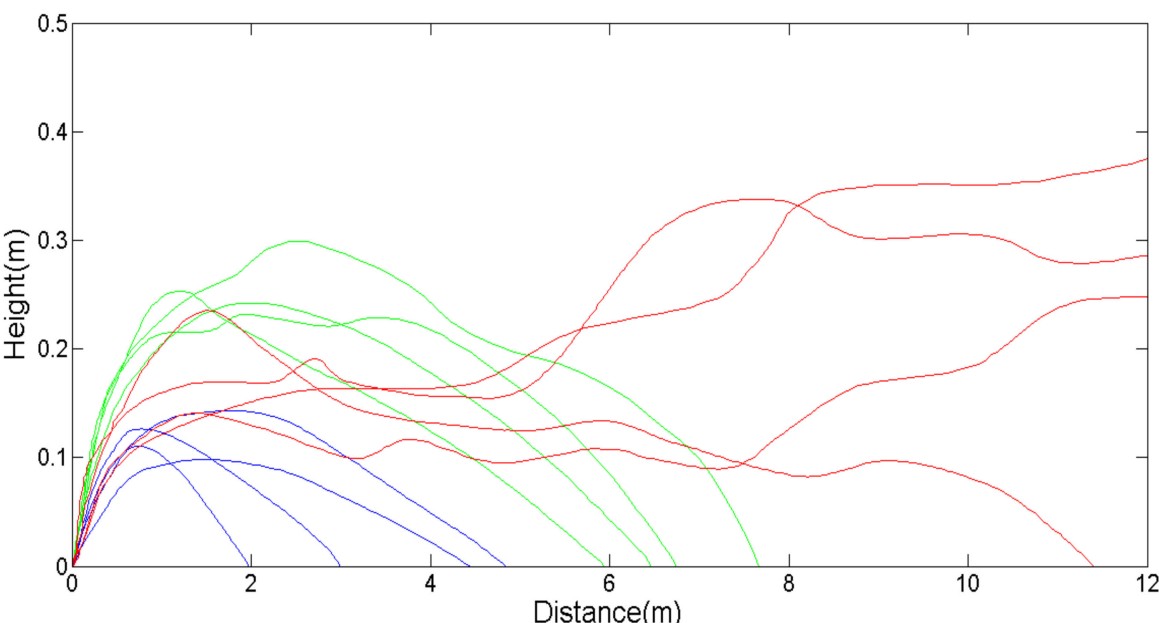

**Figure 3.** Typical two-dimensional migration trajectory of the sand particles with different sizes under the wind (red: 0.1 mm; green: 0.3 mm; blue: 0.6 mm).

The running track and migration distance of the sand particles with different sizes under the same wind velocity affects the shape of the dune. Generally, the sand particles with a size between 0.2 and 0.15 mm are most likely to jump, while those that are greater than 0.5 mm are prone to creep [59]. Therefore, after a period of migration, the sand particles of a different velocity of migration in the dune are separated naturally [5,34,60]. Previous studies showed that with the evolution of crescentic dunes, the sorting of sand particles became very regular, and the sand particles of dunes were mainly composed of fine and extremely fine sand.

To further analyze the effect of different sizes of sand particles on the morphology of dunes, we simulated the migration of dunes with the particle sizes of 0.50 mm and 0.15 mm under bi-directional winds with the wind velocity of 8 m/s. From the results of the simulation, we discovered that the dune with a particle size of 0.50 mm had a wider extended horn and a greater height (Figure 4a), while the dune with a particle size of 0.15 mm had a thinner extended horn and a smaller height (Figure 4b). This phenomenon indicated that the fine sand particles of a smaller size were more likely to drift and jump in the air by wind and were more deposited on the periphery of the dune's main body, thus increasing the sand loss of the main body. However, the coarse sand particles of a larger size were not easily carried into the air by wind because of their heavy weight, and most of them stuck to the surface of dune and exhibited creeping. Therefore, the crescentic dunes that were dominated by finer sand were more prone to single-horn extension, were thin and long in scale, and easily died out over time or might eventually be separated into several crescentic dunes of different scales (Figure 4b). In contrast, the single-horn of the crescentic dune that was dominated by coarser sand extended relatively slowly. The extension lasted for a long time and the moving distance was more regular, so the extended horn took longer to die out (Figure 4a).

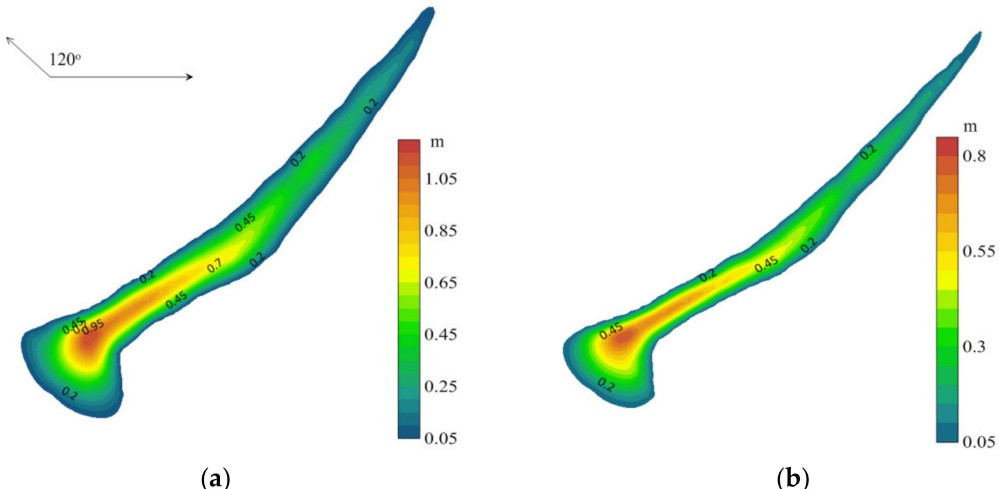

(a)                     (b)

**Figure 4.** Later evolution of the dune with different sizes of sand particles under bi-directional winds with an angle of 120° and r = 0.6 (with the same initial morphology as in Figure 2): (**a**) with the particle size of 0.50 mm; (**b**) with the particle size of 0.15 mm.

### 3.3. Simulation of the Effects of Topography on Morphological Changes in Crescentic Dunes

Although the extension and asymmetry of the two horns of crescentic dunes are mainly caused by bi-directional winds, there is also asymmetric sand flux on both sides of the main axis of crescentic dunes, which is controlled by topographic factors only under unidirectional wind [34,41]. Many crescentic dunes are located on the ground with a certain slope, which may cause the sand to slide under the action of gravity, thus causing the shape of the crescentic dune to shift and deform. Except for the slope, the size of sand particles can also have a very important effect on this action. We simulated the four final morphological changes of the crescentic dunes with the angle of θ between the different positions of dunes and the horizontal ground under a constant unidirectional wind. To eliminate the effect of the size of sand particles, we set a constant size as 0.25 mm. The simulated results are shown in Figure 5.

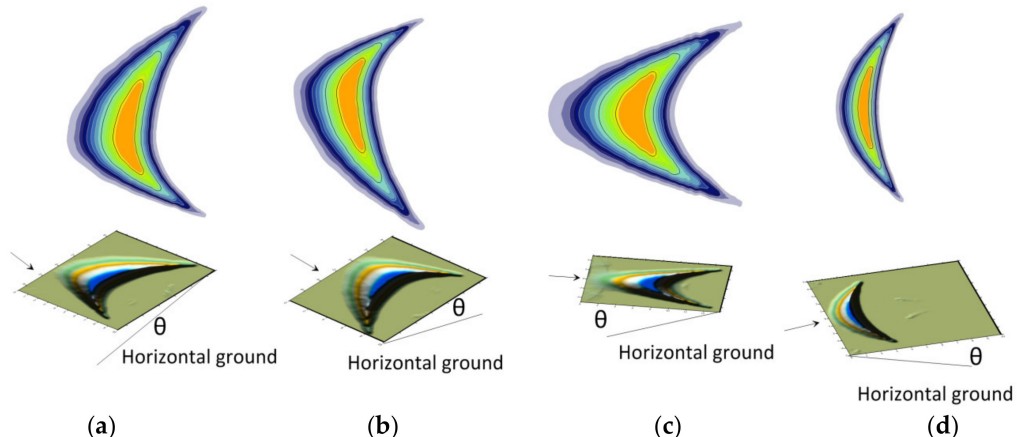

(a)           (b)           (c)           (d)

**Figure 5.** Morphological changes of crescentic dunes under different topographies. (**a**) The angle between the left horn and the horizontal ground was θ; (**b**) the angle between the right horn and the horizontal ground was θ; (**c**) the angle between the horns and the horizontal ground was θ; (**d**) the angle between the bottom of the windward toe and the horizontal ground was θ.

When the angle between the left horn of the present dune and the horizontal ground was θ, the gravity caused the sand of the left horn to slide down, and the whole dune presented the following state: the left horn was stretched, the right horn was squeezed, and the crest was shifted to the right part, forming a crescentic dune with an elongated horn

(Figure 5a). In contrast, when the angle between the right horn and the horizontal ground was θ, the situation showed the opposite pattern (Figure 5b). When the windward slope of the dune was on the top of the slope, the gravity caused the sand in the crest to slide downward, so this part was elongated and moved towards the downwind direction of the slope. The sand in the leeward toes also slipped downward under the action of the gravity. At last, the whole dune showed the state of stretching in the crest (Figure 5c). When the two horns of the dune were on the top of the slope, the gravity made the sand of the two horns slide down, so that they were compressed and shortened. The sand of the crest also slipped downward under the action of the gravity, but the position of the crest was almost unchanged. Finally, the whole dune presented a state of compression and tended to be distributed on a line (Figure 5d). From the simulation results in Figure 5, we revealed that the sand deposition that was caused by the gravity would have a different deviation in different directions under different topographic conditions, and finally the whole dune would be stretched towards the downward direction.

*3.4. Simulation of the Effects of Epiphytic Vegetation on Morphological Changes in Crescentic Dunes*

Vegetation cover on the surface of the crescentic dunes also has an extremely important influence on the dunes' morphological changes, which is mainly reflected in the fact that the vegetation can block or slow down the movement of the sand particles. Over time, different cover levels of vegetation lead to different morphological changes in the dunes [61]. In this study, we mainly used Equations (10) and (15) to simulate the evolution of the three-dimensional morphology of the dunes, and used the RANS model to generate an average 3D wind field structure. The vegetation height was set to 20 cm, and the maximum height of the dune was 10 cm. The blocking effect of vegetation on the sand particles was set to 100%. The distribution of vegetation was shown in Figure 6. We only set a vegetation distribution pattern to simulate the effect of the vegetation cover on the morphology evolution of the standard crescentic dune. Figure 6a was the initial morphology of the dune with vegetation distributed on the two horns. In the areas where there was no vegetation barrier, the sand particles migrated rapidly backward under the action of wind. First, the windward toe gradually disappeared (Figure 6b), then the dune was completely deformed. The original windward slope reversed backward and was driven to the original leeward slope by the wind (Figure 6c), or even became an inverted shape of "U" (Figure 6d). Then the dune extended parallel to the wind direction and was gradually stretched and lengthened (Figure 6e). The sand flow in the middle part still moved backward under the continuous action of the unidirectional wind, forming a longer trailing tail, and the height was gradually reduced, thus forming a shallow "V" shape (Figure 6f). More and more sand accumulated in the areas that were blocked by vegetation, forming higher and heavier sand piles centered on vegetation-blocking positions and expanding outwards and backward.

The results that were obtained by the model were only theoretical results, whereas in reality, the vegetation cover might change the turbulence of the lateral and the leeward slope, making the migration and deposition of sand particles more irregular. The sand particles swirled around the leeward slope under the influence of vortices and gradually sank to the leeward slope after bypassing the vegetation cover. In addition, the wind flow on the side of the vegetation cover was divided into the two sides, which would form the acceleration effect of wind flow on both slopes. Then, the dune's surface without vegetation would obtain a greater surface shear stress [49,62].

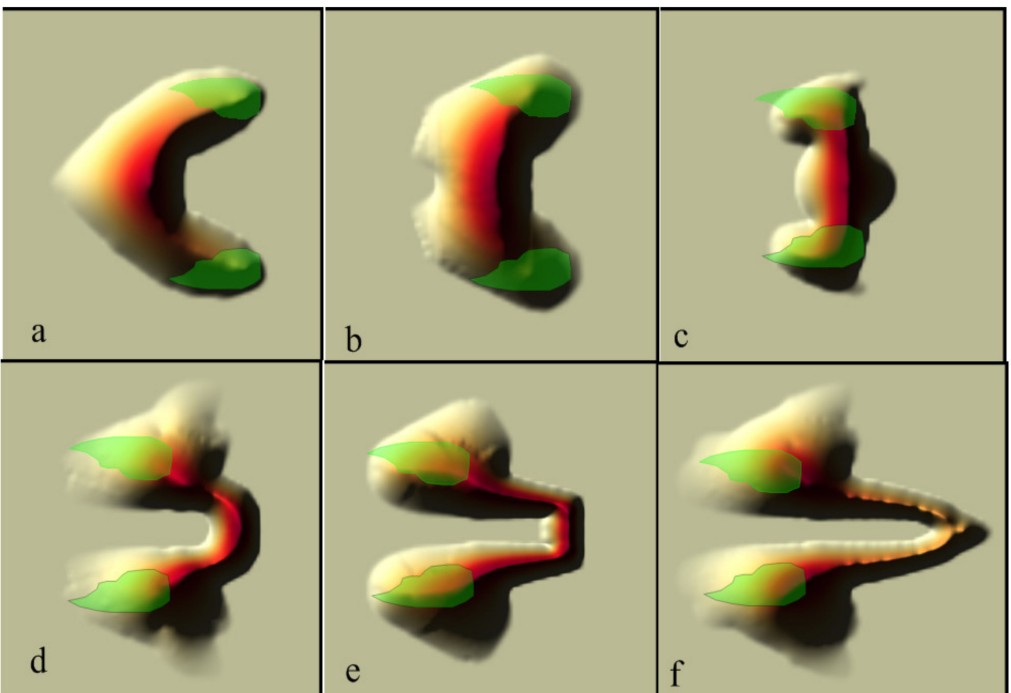

**Figure 6.** Evolution on morphology of crescentic dune under epiphytic vegetation barrier and unidirectional wind action (translucent green areas represented areas covered by vegetation). (**a**) The initial morphology of the dune with vegetation distributed on the two horns; (**b**) the two leeward toes and the windward toe disappeared; (**c**) the original windward slope moved to the position of the original leeward slope; (**d**) the dune became an inverted "U" shape; (**e**) the dune was gradually stretched and lengthened; (**f**) the dune formed a shallow "V" shape.

*3.5. Simulation of the Effects of Dune Collisions on Morphological Changes in Crescentic Dunes*

The dune collision is another important reason for the difference between the two horns of the crescentic dune. We simulated the effect of the collisions of two independent dunes on morphological changes (Figure 7). It could be seen from the simulation results that if the axes of the initial two crescentic dunes with different volumes were different, they would evolve into another two new crescentic dunes with different volumes after collision. The volume of the initial two dunes in Figure 7a was 4.21 m$^3$ and 18.27 m$^3$ and the volume ratio was 0.23, while the volume of the initial two dunes in Figure 7b was 2.98 m$^3$ and 14.38 m$^3$ and the volume ratio was 0.20. The evolved volume of the two dunes in Figure 7a was 18.9 m$^3$ and 3.58 m$^3$, respectively, while that in Figure 7b was 16.68 m$^3$ and 0.68 m$^3$, respectively. Two dunes with a similar volume ratio evolved new dunes with a large volume difference after collision, indicating that the sand flux in the downwind direction was the reason for the evolution of the dunes. The volume ratio and collision position of the two dunes determined the future evolution of the dunes. The reflux zone of the leeward slope also greatly affected the sand flux from the front dune, thus redistributing the spatial distribution of the sand particles moving downwind. It is generally believed that the larger the volume of sand dunes, the slower the velocity of movement. When two dunes collide, if the crests overlap, they converge to form a larger dune with a lower migration velocity; if the crests do not overlap, and the local sustained sand flux is interrupted, a new small dune forms. The small dune, therefore, accelerates the separation of the two dunes because of the faster movement.

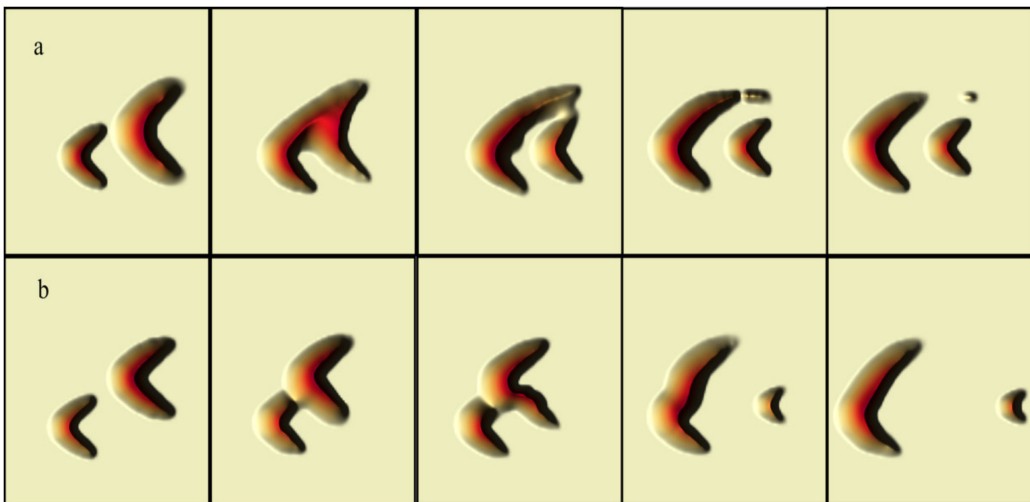

**Figure 7.** Morphological changes of crescentic dunes during collision migration. (**a**) The two dunes have a volume ratio of 0.23 and a closer distance between the axes; (**b**) the two dunes have a volume ratio of 0.2 and a longer distance between the axes.

## 4. Discussion

From our simulation, we found that the coexistence of multiple inducers led to the asymmetric characteristics of the two horns of the crescentic dunes. To further analyze the causes, the flow field on the dunes' surface was discussed, as follows.

### 4.1. Influence of Bi-Directional Winds on the Asymmetry of Sand Dunes

Figure 2 showed that the bi-directional winds at a fixed frequency, velocity, and duration could induce a different morphological evolution of the crescentic dunes. However, the different duration of the two winds would also cause the dunes to take different forms [10]. Figure 8 further simulated the flow-field structure of several dunes' morphologies in response to bi-directional winds with different angles. The simulation results revealed that two vortices existed on the leeward slope when the angles of the bi-directional winds were 30° and 60°. The flow of vortices carried lots of sand particles from the main dune to the two horns, allowing the horns to increase and extend (Figure 8(a1,2,b1,2)). There was no reverse offset of the movement in the sand particles that were driven by the bi-directional winds, meaning that the dunes maintained their crescentic shape. When the angle of the bi-directional winds was greater than or equal to 90°, the wind flow blew from the side and could not carry the sand particles from the main dune to the original extended horn. In this case, the horn no longer extended along the original direction and was prone to lateral migration under the influence of subwind. Then, the horn near the wind disappeared, while the horn far away from the wind extended. A large amount of stored sand was concentrated on another extended horn, which resulted in a significant extension of this horn (Figure 8c2–e2). This was because the winds on both sides obviously squeezed the overall amount of the dunes, meaning that they tended to be linear dunes or Seif dunes [9,63–65]. Therefore, the bidirectional-winds' angle of 90° was the threshold for the change of the dune's morphology, and the difference in the transport amount of the sand particles between the two horns inevitably led to the asymmetry of the sand dunes. From the above analysis, we found that the morphological changes of the dunes, caused by the changing wind direction, would in turn affect the flow field around the dunes, which would together determine their future morphological evolution.

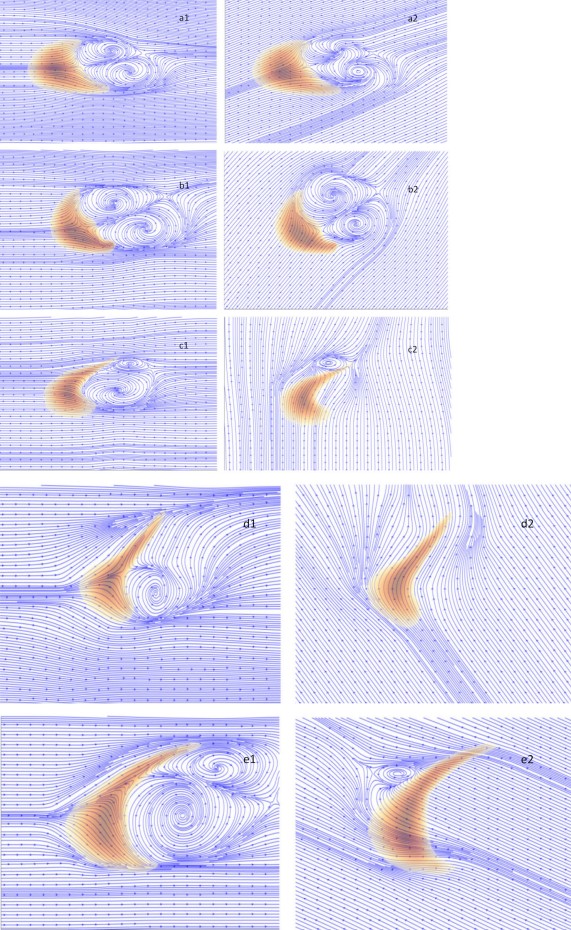

**Figure 8.** Flow field around the dune. (**a1**) Dune with morphology 1 at the direction of wind of 0°; (**a2**) dune with morphology 1, the bi-directional winds' angle of 30°; (**b1**) dune with morphology 2 at the direction of wind of 0°; (**b2**) dune with morphology 2, the bi-directional winds' angle of 60°; (**c1**) dune with morphology 3 at the direction of wind of 0°; (**c2**) dune with morphology 3, the bi-directional winds' angle of 90°; (**d1**) dune with morphology 4 at the direction of wind of 0°; (**d2**) dune with morphology 4, the bi-directional winds' angle of 120°; (**e1**) dune with morphology 5 at the direction of wind of 0°; (**e2**) dune with morphology 5, the bi-directional winds' angle of 150°. The morphology of the dune was the third subfigure of the evolution process corresponding to each angle in Figure 2. The numbers 1 and 2 behind the serial number represent different wind directions.

Our model only compared the evolution of the dunes under the influence of bi-directional winds with the same duration time; however, if the bi-directional winds were intermittent or of different order, it would cause the crescentic dunes to evolve into other forms.

### 4.2. Influence of Different Size of Sand Particles on the Asymmetry of Sand Dunes

The size of the deposited sand particles is an important factor to distinguish the sedimentary environment and reflect wind and sand dynamics. Generally, sand particles move in the form of creep and jump; however, the manner of jump can cause the sand particles to move far away, thus leading to the dispersed distribution of the deposited sand particles. Crescentic dunes have a good sorting ability for sand particles [58] and, usually, sand particles that are distributed on the crest and both sides of the windward slope are more active. From our simulation results (Figures 3 and 4), it was certain that the small sand particles had a long migration distance, fast movement, and large deposition range [66]. Thus, sand particles of a smaller size were more easily distributed along the extended horn. When the extended horn lost the sand supply from the main dune, it easily evolved into smaller crescentic dunes under the action of the continuous winds, as shown

in Figure 9. On the contrary, the morphology of the sand dune with larger sand particles varied little over time, with its sediments being concentrated, and its horn extending more strongly (Figure 4). Therefore, different sizes of sand particles have certain impacts on the asymmetry of crescentic dunes [67,68]. The coarser the sand particles, the more concentrated the dune accumulation and migration.

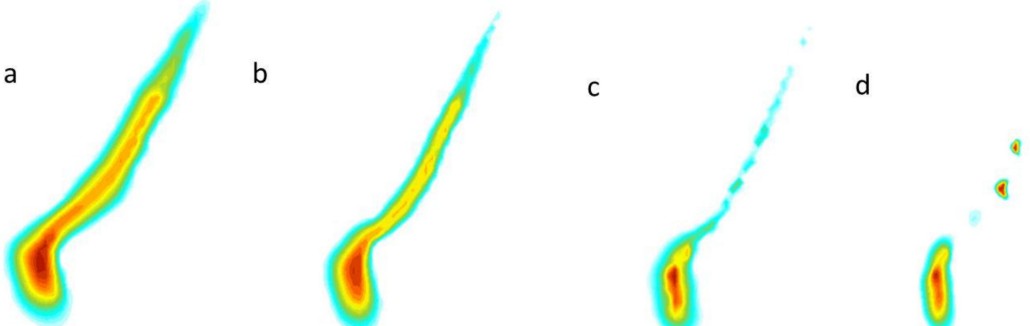

**Figure 9.** Evolution process of the break-out of extended horn. (**a**) There was enough sand supply in the extended horn to produce displacement; (**b**) the transportable sand supply became less abundant as the sand particles gradually dispersed; (**c**) the extended horn broke and a new small dune formed in the tail; (**d**) more small dunes emerged after the extended horn.

To further clarify the effect of different sizes of sand particles on the morphological evolution of crescentic dunes, we simulated the flow field and evolutions of two standard crescentic dunes with horns of the same size (Figure 10a, 0.25 mm) and different sizes of sand particles (Figure 10b, with the size of 0.20 mm in the left horn and 0.25 mm in the right horn) under unidirectional wind. We found that the sizes of the vortices that formed behind the leeward slope were consistent when the sizes of the sand particles were the same between the two horns (Figure 10a), while the sizes of the vortices would change greatly when those between the two horns were different, even if the difference was very small (Figure 10b). This was because the smaller sand particles moved faster and farther away, so those on the left horn moved further after being rolled up by the vortices, and the range of deposition was larger. Thus, the left horn of the dune extended faster (Figure 10b). In turn, the deposition form of the dune would also affect the sizes of the vortices. Therefore, the size of the sand particles can cause the different extension of the two horns of the crescentic dunes.

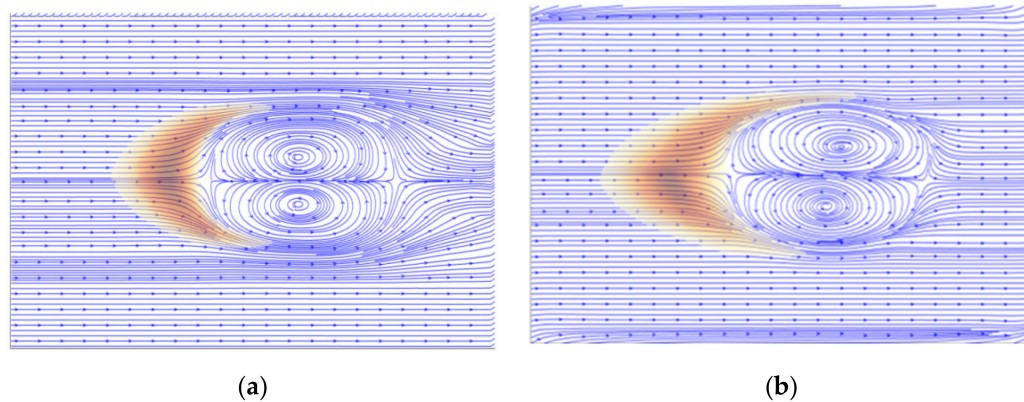

| (**a**) | (**b**) |
|---|---|

**Figure 10.** Flow field around the sand dunes (**a**) with the same sizes of sand particles on the two horns (0.25 mm); (**b**) with the different sizes of sand particles on the two horns (0.25 mm on the left horn and 0.20 mm on the right horn).

*4.3. Influence of Topography on the Asymmetry of Sand Dunes*

The influence of topography on crescentic dunes mainly manifests in the sliding of sand particles under the action of gravity [69]. Since the action of wind flow reduces the

friction between the sand particles and the dune's surface, the morphological changes of crescentic dunes under the topography are actually the result of the joint action of the wind and the gravity [70]. Generally, the horn at the lower part of the slope easily deposits the sediment from the main dunes and slides down under the gravity, and thus can easily become long and narrow, while the horn nears the upper part of the slope becomes wider because the gravity reduces the sediment and presses the remaining volume.

We simulated the final morphological changes of several different types of crescentic dunes at different slopes under the action of unidirectional wind (Figure 11). The simulation results revealed that the dunes with different heights located on the same slope varied greatly in response to the gravity, especially as the higher (thicker) the dunes, the more significant the deformation caused by the gravity. In addition, the evolution of the morphology of dunes on the slope is closely related to the location between the slope and the wind direction. When these are in the opposite direction, the climbing dunes appear.

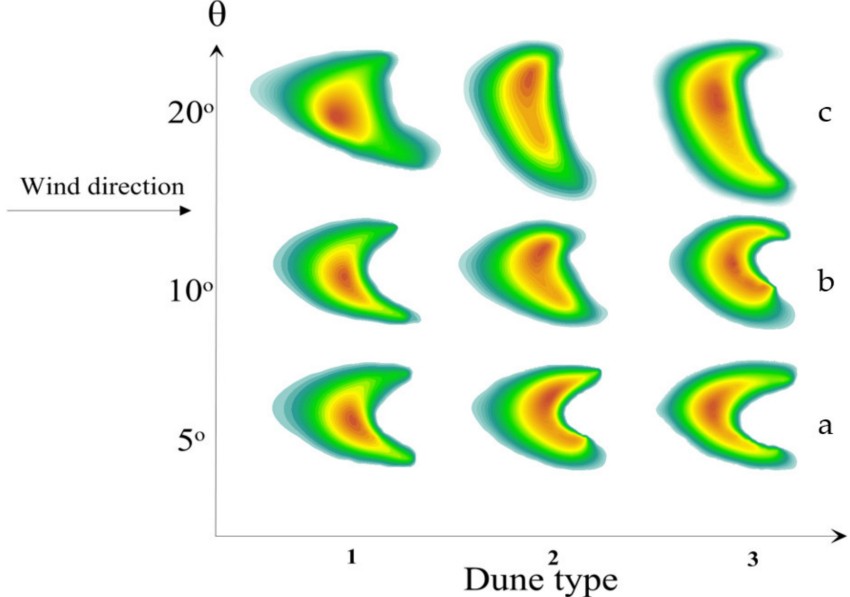

**Figure 11.** Differences in morphology of several types of crescentic dunes (the initial dune with the thickest right horn or thickest left horn or thickest middle part) affected by different slopes in the same orientation (with the angles between the right horn and the horizontal ground) under the action of unidirectional wind: (**a**) at an angle of 5°; (**b**) at an angle of 10°; (**c**) at an angle of 20°.

There were several different types of simulated crescentic dunes in Figure 11, with the thickest right horn (dunes in the first vertical column); the thickest left horn (dunes in the second vertical column); and the thickest central ridge (dunes in the third vertical column). When the angle between the right horn and the horizontal ground was 5° (Figure 11a), the right horn was almost not elongated because of the small angle of inclination and the very weak downward division force of the gravity. As a result, the dune with the thickest right horn remained unchanged (see the first dune in Figure 11a), and the dune with the thickest left horn also maintained the thickest left horn, but the thickest part slightly moved toward the central ridge and right horn (see the second dune in Figure 11a). The dune with the thickest central ridge remained the thickest, but the thickest part slightly moved to the right horn (see the third dune in Figure 11a). When the angle was 10° (Figure 11b), the downward division force of the gravity was also weak because of the small angle of inclination, causing the sand of the right horn to slide downward. Thus, the right horns of all types of crescentic dunes were slightly elongated, the thickest parts of the dunes were inclined to the right part, and the deformations of the dunes became larger. When the angle was 20° (Figure 11c), the gravity made the sand of the right horn slide down because of the large angle of inclination, so the right horn was obviously elongated, and the thickest parts

were also more inclined to the right part. Then, the whole dunes were severely deformed, showing the state of the right horn stretching and the left horn squeezing to the central axis. Therefore, the deviation of the crescentic dunes in the same orientation was related to the angles between the dunes and the horizontal ground and the morphology of the original dunes. Between them, the influence of the slope (the angle between the dune and the horizontal ground) was greatest.

### 4.4. Influence of Epiphytic Vegetation on the Asymmetry of Sand Dunes

To analyze the effect of epiphytic vegetation on the evolution of crescentic dunes, we simulated two immediate dynamics of flow fields during the morphological evolution of the dunes under vegetation cover (Figure 12). The vegetation coverage position was consistent with that of Figure 6. As shown in Figure 12, the wind flow in the central area of the dune remained unchanged, where the sand here moved backward under the action of a continuous wind flow, and then the original windward slope was retracted to the leeward slope (Figure 12a). Blocked by the epiphytic vegetation at the toes of two horns, the wind flow could only extrude and rotate to the outside of the dune, and thus the wind was stronger here. With the evolution of the sand dune, the strong wind flow squeezed into the depression zone after the leeward slope, forming two symmetrical horizontal vortices after the areas that were blocked by the vegetation (Figure 12b). Under the combination of the central wind and the two vortices of the leeward slope, two linear dunes gradually formed in the central part, and the two originally adjacent vortices were gradually pushed away by them (Figure 12b). From this, it could be seen that epiphytic vegetation was another important factor affecting the asymmetry of crescentic dunes [71,72]. However, our simulation was an ideal state of the symmetric distribution of epiphytic vegetation on the horns of the crescentic dunes. In fact, epiphytic vegetation on the desert cannot be symmetrically distributed as in the model or completely block sand's movement [73,74], so it is more likely to form an asymmetric dune.

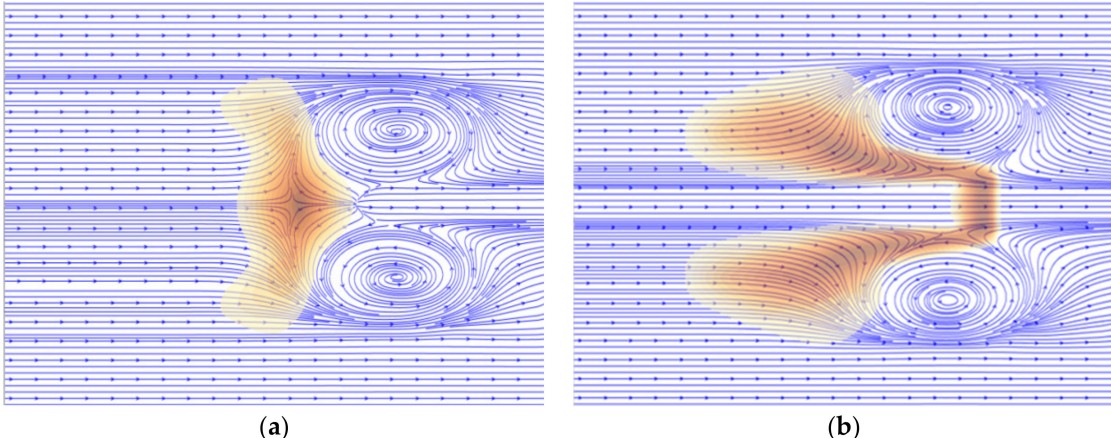

(**a**)       (**b**)

**Figure 12.** Distribution of flow fields during the evolution of sand dunes under vegetation obstruction: (**a**) corresponding to the flow field of Figure 6c; (**b**) corresponding to the flow field of Figure 6e.

### 4.5. Influence of Collision of Dunes on the Asymmetry of Sand Dunes

The differences between the horns of crescentic dunes caused by dune collision is the most common deformation phenomena in desert [4,75]. With the differences of wind velocity caused by the height of dunes and the local topography, the migration distance of sand particles also changes. Combined with the collision, the heights and shapes of the dunes change, which results in the changes of sand flux in space and time, and the uneven migration of sand in the location of the horns [4]. In general, small sand dunes often act as the active participants in dune collisions due to their fast-moving velocity. According to Parteli's study [10], the smaller the volume ratio of the dunes before the collision, the more likely the two small dunes are to merge into a large dune. When the volume ratio

of the two dunes is constant before the collision, the larger the deflection angle of the central axis of the two dunes, the greater the volume difference between the two new dunes after the collision, and the larger crescentic dune that is formed is more likely to have asymmetric horns.

To further analyze how the changes in the dunes' morphology occurred after the collision, we simulated the changes of flow field on the dunes' surface before and after the collision (Figure 13). Before the collision, the two dunes were separated, and the extended horn below the larger crescentic dune was greatly affected by the vortex behind the leeward slope (Figure 13a). After the collision, an additional force was added to the outside of this extended horn, which came from the vortex behind the leeward slope of the original smaller crescentic dune. Thus, the separation of the extended horn from the main dune was accelerated by the combination of the two vortices and the main wind. In addition, due to the increased volume of the new dune that merged after the collision, the sand flow in the middle of the dune could not be continuously supplied to the extended horn. Moreover, the volume of the extended horn was smaller than the main body of the front dune, so it moved rapidly under the main wind and accelerated its separation from the main dune. Then, a small crescentic dune appeared behind the main dune (see the third stages in Figure 7). It could be seen that the two dunes would connect into a new dune after the collision, changing the force of the flow field around the original single dune. Under the action of the new flow field, the morphology of the dunes changed greatly, and the two horns also presented an obvious asymmetry.

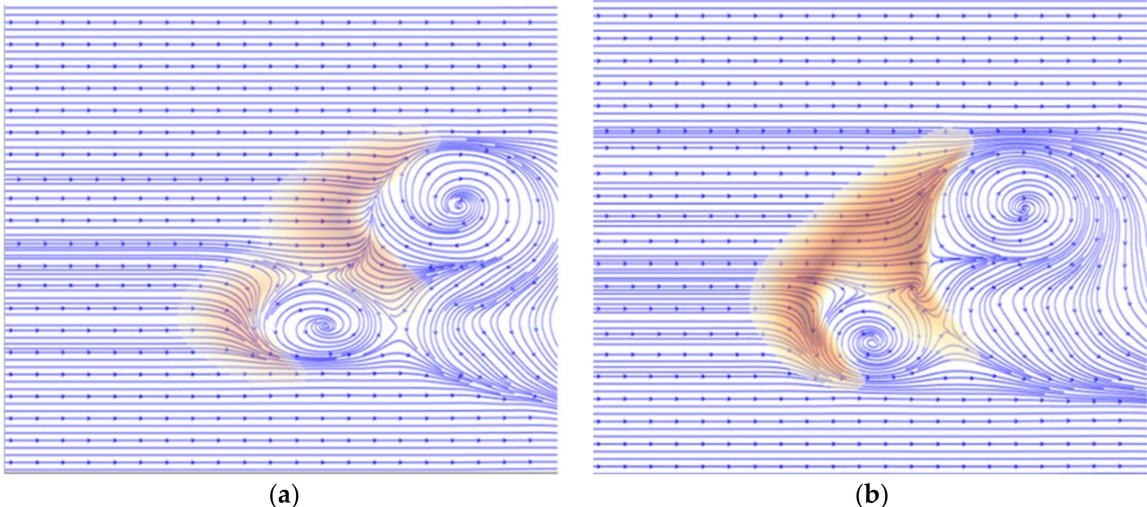

(**a**)   (**b**)

**Figure 13.** Changes of surface flow field before and after the collision of the two crescentic sand dunes: (**a**) corresponding to the second stage in Figure 7b; (**b**) corresponding to the second stage in Figure 7a.

### 4.6. Erosion and Mechanism of Asymmetric Crescentic Sand Dunes on the Environment

The crescentic sand dunes move fast and can maintain this speed even when the sand supply is insufficient, so they can pose a threat to the surrounding environment through rapid sand erosion. The mechanism of action is as follows: when the wind blows over the sand dunes, the wind field from the windward slope to the leeward slope shows a concave streamline. With the increasing curvature, the spatial variation of the wind velocity in the flow field increases. Due to the obstruction by the windward slope, the velocity of the wind flow reduces, but it increases the handling of sand particles on the windward slope, so the windward slope presents the characteristics of low wind velocity and high sand flux—that is, the amount of wind erosion on the windward slope is greater than the amount of sand deposition, thus showing the erosion of the wind and sand environment. Asymmetrical crescentic dunes with single horn extension increase the area of the windward slope

(Figure 2), which undoubtedly increases the spatial erosion of the windward slope, so the asymmetric crescentic dunes are more able to erode the environment.

In addition, the wind flow passes and accelerates along the toes of the windward slope towards the toes of the two horns, carrying the sand particles downward, causing the erosion on the horns, and moving the dunes forward. At the same time, the surface sediments on both sides of the dunes are also affected by the accelerated wind flow, forming the areas of wind erosion with a large erosion degree. When the sand supply is sufficient, the wind–sand flow in front of the dune can easily reach a saturated state. The saturated wind–sand flow is hindered when encountering the two horns, the wind velocity decreases, and the sand particles deposit, so the sand particles that are carried by wind are moved to the two horns. However, when the sand supply is insufficient, the wind–sand flow is unsaturated, and it is easier to carry the surface sediment when passing through the horns, meaning that its wind erosion ability is enhanced. Larger asymmetric crescentic dunes may develop many small crescentic dunes in the downwind based on this principle (Figure 9). The formation of these small dunes not only increases the distribution area of the crescentic dunes, but they also move faster and are more able to erode the environment.

Moreover, the vortices after the leeward slope of the crescentic dune can block the sand flux to the downwind direction, causing the wind–sand flow to adopt a high unsaturated state. Therefore, the vortices can carry more sand particles on the ground or two horns after the leeward slope, strengthening the erosion effect. However, the asymmetric crescentic dunes can form various different vortices after the leeward slope (Figure 8), which may change the range of spatial erosion by the vortices on the sand particles to varying degrees. In particular, the crescentic dune with a single horn extended has a larger range of leeward-slope shielding, which greatly increases the amount of sand flux at local locations.

In a word, asymmetric crescentic dunes, due to their diverse shapes and low concentration of sand, are easier to be disassembled and separated by winds from all directions, and then form many small dunes. Then, the erosion of wind–sand flow changes from a single position to multiple positions. The constant changes of surface flow field, movement direction of sand particles, amount of sand flux, position of erosion, and mode of erosion can all cause misjudgment to the area of sand erosion at the downwind and increase the difficulty of the defense of wind and sand erosion.

## 5. Conclusions

In this study, the causes of the asymmetry of crescentic dunes were explored and the formation and evolution of the asymmetric crescentic dune were simulated.

Among the influencing factors, angle and frequency between the bi-directional winds, the size of sand particles, topography, epiphytic vegetation, and collision of dunes were the main reasons for the asymmetric evolution of the two horns of the crescentic dunes.

The analysis of the bi-directional winds suggested that the duration, proportion of duration, and the angle of the bi-directional winds all affected the morphological characteristics of the dunes. An angle of 90° was a threshold for a larger extension of a single horn. In addition, no stable and symmetrical vortices existed behind the leeward slope under the action of the bi-directional winds, which was also the most important reason for the asymmetric evolution of the crescentic dunes. The analysis of the size of the sand particles revealed that sand particles of a smaller size were easy to jump, so a crescentic dune with finer sand was more prone to a single horn extension or fracture or extinction under the action of wind. On the contrary, the continuity of sediment deposition of the crescentic dune with coarser sand was better, and the extension of the single horn lasted longer.

The influence of topography on the morphology of the crescentic dune was mainly reflected in different directions and different degrees of deviation under different terrain conditions caused by the gravity. The range of extension was determined by the angle between the plane where the dune was located and the horizontal ground. Slope and gravity dominate the direction of extension and the degree of offset of the two horns.

Epiphytic vegetation on the dune hindered the movement of the sand particles and had a great influence on the asymmetric deformation of the crescentic dune. Under the action of unidirectional wind, the areas of the dunes that were not covered by vegetation migrated fast, which could increase the asymmetric evolution of the crescentic dunes.

The structure of vortices changed significantly before and after the dune collision, and the resulting spatial distribution in the sand flux was another reason for the change in the morphology of the dune. Moreover, the further the distance between the main axes of the two dunes before the collision, the more significant the asymmetry of the dunes after the collision.

In conclusion, all the influencing factors of the asymmetry of sand dunes indicated that the inter influence between the flow field on the dune's surface and the dune's morphology was the reason for the further evolution of the dune's morphology. Our study simulated the structure of flow field under various morphologies of dunes, providing a reliable explanation for the various morphological evolutions of sand dunes under the fluctuating wind field.

**Author Contributions:** H.Z. performed the numerical simulations and wrote this article. Z.W. revised the manuscript, C.L., J.Z., Z.Z., J.H., L.C., L.S., J.M. and B.X. took part in the field investigation. All authors have read and agreed to the published version of the manuscript.

**Funding:** This study was supported by the Basic Scientific research Foundation of China Earthquake Administration (2020IESLZ04); the Science for Earthquake Resilience of China Earthquake Administration (XH20059); the National Natural Science Foundation of China (Nos. 31760125, 41761006); the Key Projects of Gansu Natural Science Foundation (20JR5RA097, 20JR5RA093); and the Key R & D Program of Gansu Province (No. 20YF8FA105).

**Institutional Review Board Statement:** Not applicable.

**Informed Consent Statement:** Not applicable.

**Data Availability Statement:** Not applicable.

**Conflicts of Interest:** The authors declare no competing interest.

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
