# Peer review of "Numerical Simulation Analysis of the Formation and Morphological Evolution of Asymmetric Crescentic Dunes"

_sustainability, doi:10.3390/su14148966_

Round 1

Reviewer 1 Report

I will start with an apology to the authors about my tardiness of this review, this time of year was particularly busy for me.

This manuscript, entitled ''Numerical simulation analysis of the formation and morphological evolution of asymmetric crescentic dunes''by Zhang Huiwen Li Changlong, Zhang Jianhui, Wu Zhen, Zhang Zhiping, Hu Jing, Cao Lei, Song Longlong, Ma Jianping, Xiao Binreports a series of numerical simulations based on a Reynolds-Average Navier–Stokes and mass balance model to simulate the asymmetric shape of crescentic dunes and explore the cause of this asymmetry by taking into accountdifferents parameters, as the effect of bi-directional winds, the grain-size of sand particlesthe irregularity of topography, the presence of vegetation. As a result, the authors observe that the angle of bi-directional winds controls the structure of the vortexesdeveloped around the dune. The extension of the dune horns is enhanced with coarsest materialsThe asymmetric shape is more easily developed when the topography presents irregularities while the dune migration is influenced by the presence of anepiphytic vegetation on the dune surface. Moreover, the way the dunes interact plays a role on their final shapes.

To my opinion, this manuscript presents numerous original and interesting qualitative results that expose the development of asymmetric dunes, commonly observed in natural landscapes that justifies the interest of the study and its publication in MDPI. I particularly like the complete exploration of the different key factors in the develoment of their asymmetric shapeHowever, the present manuscript requires some improvements before publication that I will suggest below.

1. The introduction lacks of a more detailed and deep presentation of pre-existing models, experiments, and numerical simulations of asymmetric dunes. The problematic that emerges from the community and the choice of the RANS and mass model made in this study are not enough justified and clearly exposed. As numerous factors are explored and tested hereI suggest to first present a general overview of the main factors that need to be explored, the different models used (with their interets and limits) and the consequent results obtained in the literature, and to better justify your contribution through an organized presentation of the different simulations made and the choice of the model. To my opinion, it would be appreciated by the readers to underline the qualitative aspect of your exploration.

2. Concerning the effects of the particles size on the mohplogical evolution of the dune, I can clearly observe on Figure 3, the different two-dimentional trajectories of the particles submitted to a wind of same velocity when the particle size is modified. However, with time the late evolution of the dune seems to be approximately the same. Please, can you explain why ? 

3. The exploration of the presence of vegetation on the surface of the dune is not enough clear for me. Please, could you explain in more details the physical description made in the model for this aspect ? And justify the choice of a vegetation layer deeper than the dune thickness ? Moreover, I would like to understand why this effect drives to a U-shape dune ?

4. There are problems with figures 8, 10, 12, 13 in the part discussion that makes difficult the reading of this part. Please, you need to correct this for the next proposition.

5. Since the authors have made the choice to present a large number of results, the discussion and conclusion parts may benefit from a more well organized presentation by defining more clearly the principal messages that need to be underlined for the community and in response to the problematic that will be clearly exposed in introduction. Please, do not hesitate to underline the qualitative nature of your exploration. 

When these modifications will be made, the paper should be greatly improved. 

Author Response

Dear reviewer: We appreciate the reviewers for their valuable comments, which is helpful to improve the quality of our present manuscript. According to the comments and suggestions, we have revised our manuscript one by one. Specific modifications are included in the attached document.

Reviewer 2 Report

Line 44: This is very important because the final goal of this study is to identify wind erosion processes that can be used to reduce the effects and damages of this phenomenon. Therefore, more explanation is needed in the introduction

Line 144: The wind is not able to pick up sand particles at a certain speed, so it is need to determine the erosion threshold speed at which the wind can move the particles.

Line 184: if the force of jumping sand in the air flow has the same condition to the same motion of the fluid?

Line 355-358: Where and how did this simulation take place? What was the duration of the changes in Barchans shape? What has been the vegetation in terms of density and infiltration? For example, does the simulated vegetation with 30% permeability with 60 or 70% permeability have the same conditions in the morphological change of the sand dune? Further explanation is needed.

Line 500: if the conditions of formation and morphological changes in the hills you are interested in the same with a height of 20-37 cm with larger crescent hills up to height of 5 meters (which are known as active dune hills in wind erosion) is the same?

What can help identify these factors affecting the wind erosion process and the damage caused by these sand crescents? Certainly, if this issue is further highlighted, it will show the importance of the article, especially in the Minqin area, where wind erosion and the movement of sands dune cause a lot of damage in this region every year.

In the results, discussion and conclusions, the same content is repeated, so the volume and number of pages of the article has increased. Please do a brief review if possible.

Author Response

Dear reviewers: We appreciate the reviewers for their valuable comments, which is helpful to improve the quality of our present manuscript. According to the comments and suggestions, we have revised our manuscript one by one. Specific modifications are included in the attached document.

Reviewer 3 Report

This research paper (Sustainability-1759189-peer-review-v1) deals with the study of numerical simulation on the morphological evolution of asymmetric crescentic dunes in the Gobi Desert area of China. This paper presented systematic methodologies and physical factors for the development of empirical models on the development of asymmetric crescentic dunes. The paper is one of the pioneer contributions in understanding the morphological evolution of sand dunes, which is most suitable for the publication in Sustanability. However, there are certain criticisms observed in this article which needs further revision.

Abstract: There are significant lacks of clarity with respect to the scientific objective of the present study. It is not properly defined in this section. This has to be re-written before publication.

Some issues need to be clarified as follows:

1.  What is the importance of sand particle size in the evolution of sand dune in the Gobi Desert? This factor is essential in understanding the morphology of dunes and should be discussed in detail.

2.  What is the dimension of the sand dunes in the Gobi Desert? Provide some statistical data to validate your numerical model.

Research methodologies and their parameters are systematically organized.

There are numerous grammatical corrections and re-organization of sentences is commented on the main article in the form of annotation. That all points should be considered before submission of your revised version of the paper.

I feel this manuscript should be appropriate to recommended for publication in the Sustanability with major revision

Author Response

(The authors gave the same response as above.)

Round 2

Reviewer 1 Report

The authors have responded well, thoughtfully, and in detail to my first review. For the most part, I accept the authors’ answers to my comments.